# Stem Cells: Current Status and Therapeutic Implications

**DOI:** 10.3390/genes11111372

**Published:** 2020-11-20

**Authors:** Kaladhar B. Reddy

**Affiliations:** 1Department of Pathology, Wayne State University, Detroit, MI 48201, USA; kreddy@med.wayne.edu; 2Karmanos Cancer Institute, Wayne State University, Detroit, MI 48201, USA

**Keywords:** cancer stem cells, stem cell niche, cancer

## Abstract

Cancer stem cells (CSCs) are a class of pluripotent cells that have been observed in most types of cancers. Evolving evidence suggests that CSCs, has the ability to self-renew and initiate tumors, may be responsible for promoting therapeutic resistance, tumor recurrence and metastasis. Tumor heterogeneity is originating from CSCs and its progenitors are recognized as major difficulty in efficaciously treating cancer patients. Therefore, understanding the biological mechanisms by which CSCs survive chemo- and-radiation therapy has the potential to identify new therapeutic strategies in the future. In this review, we summarized recent advances in CSC biology and their environment, and discuss about the potential therapies to prevent therapeutic resistance.

## 1. Introduction

Stem cells are a small number of pluripotent cells in tissue that can either mitotically divide to self-renew and produce more stem cells or differentiate into mature cells of a particular tissue. There are two types of stem cells-embryonic stem cells (ESC) and adult stem cells. ESC is obtained from a 3–5 day-old blastocyst, and is capable of giving rise to any type of organ in the body [1]. Adult stem cells are restricted to a specific tissue and have the ability to self-renewal and produce mature cells under highly controlled microenvironment [2]. Adult stem cells have two characteristic features. First, they can self-replicate for long periods of time. Second, they give can rise to mature cell types that have characteristic morphologies (shapes) and specialized functions. Normally, adult stem cells generate an intermediate cell called a precursor cell. Precursor cells are usually regarded as tissue-specific stem cells that are committed to differentiate along a particular cellular development pathway [3]. Until recently, it was believed that adult stem cells could create only similar types of cells. For instance, it was thought that stem cells residing in the bone marrow could give rise only to blood cells. However, new data suggests that adult stem cells are able to create unrelated types of cells. For instance, bone marrow stem cells may be able to create muscle cells or β islet cells [4,5]. Their primary functions are to maintain the steady-state functioning of a cell—called homeostasis—and, with limitations, to replace cells that die because of injury or disease.

## 2. Key Features of Normal and Cancer Stem Cells

There is evidence to show both normal stem cells (NSCs) and cancer stem cells (CSCs) have many similarities, including migratory, self-renewal, slow cycling and differentiation properties [6]. Both NSCs and CSCs have the capacity for asymmetric division for self-renewal, which produces stem cells and progenitor cells, which play a major role in tissue repair or cancer. They both use similar signaling pathways (Wnt, Notch, Sonic Hedgehog, etc.) for self-renewal [7,8]. In both, life span is extended by telomeres and telomerase activity [9], and they can be identified based on cell-surface markers [10]. Both NSCs and CSCs escapes immune surveillance by reducing the expression of M1 macrophage inhibitors CD200 and CD44 blocking macrophage M2 polarization and phagocytic activity. In addition, tumor microenvironment (TME), like IL4, IL-6, IL-10, TGF-β, paralyzing the immune responses [11]. Some of the differences between NSCs and CSCs are: NSCs have extensive self-renewal capacity, highly regulated self-renewal and differentiation, normal karyotype, quiescent, and can generate normal progeny with limited proliferative potential. CSCs have indefinite self-renewal capacity, highly dysregulated self-renewal and differentiation, abnormal karyotype [12,13], mitotically less active than other cancer cells and have the capacity to produce phenotypically diverse progeny. CSCs are highly resistant to lack of oxygen compared to NSCs [14]. NSCs use oxidative phosphorylation (OXPHOS) as a primary source of energy, whereas glycolysis as a main source of energy [15,16]. One of the major differences between NSCs and CSCs is their degree of dependence on the stem cell niche. NSC is supported by niche to maintain homeostasis, whereas, CSCs play a major role in deregulation of the niche by promoting invasion and metastasis [17,18] (Table 1).

## 3. Identification of Cancer Stem Cells

Proportion of CSCs is low compared to total mass of the tumor(s), cell-surface markers have proven useful for isolation and enriching CSCs from different cancers (Table 2). For the first time, it was shown CD34^+^CD38^−^ stem cells initiated human myeloid leukemia after transplantation into SCID mice [19]. Breast cancer stem cells identified by CD44^+^CD24^−^ cells formed tumors into Nod/Scid mice [20]. CD44 is a transmembrane glycoprotein on the surface of endothelial cells and leukocytes, which binds to extracellular matrix and activates EGFR and ErbB2 and enhances cell migration and differentiation [21,22]. CSC cell surface marker CD44 is used as a diagnostic marker for identification in breast, head and neck, prostate, lung, hepatocellular, pancreatic and squamous-cell carcinoma [23,24]. CD133 (Prominin-1) was originally described as a CSC marker for glioblastoma [25]. Moreover, glioblastoma tumors in vivo have shown that only the CD133^+^ cells had the ability to maintain tumorigenesis and generate heterogeneity [26]. In several cancers, including breast cancer, CD44 and CD24 cell surface markers have been used to isolate CSCs; however, they should not be regarded as universal markers. A number of studies have shown that CSCs were found in both the CD44^+^ CD24^−^ and CD44^+^CD24^+^ fractions [27]. A similar story holds true for colorectal cancer in which the EpCAM^hi^CD44^+^ CSC subpopulation shared small overlap with CD133 [28]. However, in pancreatic cancer, where overlap between the CD133^+^ and CD44^+^CD24^+^ populations varied significantly between specimens [29]. One of the reasons for limited overlap between the phenotypes of CSCs isolated from same tumor may be because of the presence of multiple CSC pools or variations arising from different isolation techniques. In addition, stringent assays to prove self-renewing activity was not applied in some cases. Identifying CSCs by using the expression of markers is the most popular method today, hence researchers are trying to identify specific markers for CSCs.

## 4. Immunological Characteristics of Cancer Stem Cells

CSCs have been shown to have immunosuppressive effects like stem cells [30,31]. Previous study of normal mesenchymal, haemopoietic stem cells was shown to interfere with T cell functions by impairing IFN-γ production or by releasing large amounts of IL-10 [32,33]. Secretion of interleukins (IL), such as IL-4, IL-10, IL-13, and TGF-β by CSCs was shown to have immunosuppressive effects on T cells, antigen-presenting cells, and natural killer (NK) cells [34]. In the aggressive melanoma cell line A375, it was shown that the immunogenic tumor-associated antigen, melanoma antigen recognized by T cells (MART-1) was expressed on differentiated melanoma cells, but not on malignant melanoma initiating cells (MMICs). Therefore, MART-1 specific T cells cannot eliminate MMICs [35]. Similarly, CD44^high^/CD24^low^ breast CSCs selectively escape from NK cell mediated killing and trastuzumab-dependent ADCC [36]. The expression of major histocompatibility complex I (MHC I) is often lower on the surface, and CSCs are more likely to be susceptible to NK cell-mediated cytotoxicity. However, in brain and breast cancers, CSCs often have a deficiency in NK-activating ligands, like NKG2D [37]. In addition, breast CSCs and glioblastoma stem cells secrete more TGF-β as compared to normal tumor cells [38,39]. Colon CSCs secrete higher levels of interleukin 4, which promotes drug resistance and inhibits anti-tumor immune responses [38,40]. This data suggests both NSCs and CSCs has the ability to regulate diverse membrane-bound and soluble factors, which enable these cells to modulate immune responses and protect them against immune-mediated destruction. The question whether or not CSCs are capable of seeding tumors depends on their ability to escape the immune system. However, most of the studies were done in mouse models lacking functional T, B, and NK cells, such as the NSG mice [41,42]. While studies are done in a completely immunodeficient animal model, however, for translational research, this model may not be ideal, as human subjects are immune-competent. The role of CSCs in tumor progression and tumor immune escape is best tested in models possessing a functional immune system.

## 5. Cancer Stem Cell’s Niche

Niches are specialized microenvironments that regulate adult stem cells through cell-cell contacts and secreted factors. Normal niches are comprised of fibroblastic cells, endothelial and perivascular cells, extracellular matrix (ECM) components, immune cells, network of cytokines and growth factors [43]. Niches are a physical anchoring site for stem cells [17]. The niche maintains stem cells primarily in a quiescent state by providing signals that inhibit cell proliferation and growth as shown by stem cell’s ability to retain bromodeoxyuridine labeling for a relatively long period of time in the hematopoietic, intestinal and hair follicle system [44,45,46]. The relationship between CSCs and ECM seems to be bidirectional in most cancers: the niche has the ability to alter the cellular fate of cancer cells, and conversely; CSCs can alter their microenvironment [47,48,49]. Previous studies have shown that CSCs and endothelial cells in the tumor microenvironment can transform normal fibroblasts into cancer-associated fibroblasts (CAFs) [50]. CAFs stimulate stemness via activation of Wnt and Notch pathways. Previous studies have shown that Wnt over expression and activation leads to leads to expression of stem cell markers in epithelial cells [51,52]. Mesenchymal stem cells are multipotent stromal cells that have been implicated in restoration of CSCs, as they secrete a variety of cytokines that have both autocrine and paracrine functions in the tumor milieu. Mesenchymal stem cells can promote cancer stemness through the NF-kB pathway by secreting CXCL12, interleukin-6(IL-6) and IL-8 [53]. To evade immune surveillance, the niche must immunosuppress the cytotoxic function and infiltration of natural killer cells (NKs) and CD8^+^T cells. It was recently shown a sub-population of anti-tumor CD103^+^ dendritic cells, which can efficiently stimulate CD8^+^T cells, is masked from tumor antigens by other tolerating antigen-presenting myeloid cell populations [54]. Tumor-associated macrophages secrete Transforming growth factor β (TFG-β), which recruits T regulatory cells that also participate in immunosuppression [55]. Hypoxic CSCs impede CD8+T cell proliferation and activation and inhibit immunosurveillance [56]. Hypoxia promotes CSC survival through the ROS-induced TGF-β pathway. Activation of TGF-β as well as Wnt signaling pathways induces stemness by promoting an undifferentiated state in tumor cells [57,58]. In addition, normal stem cell niches, anchoring stem cells to the niche through cell-cell contacts, are critical for self-renewal. For example, Notch and Hedgehog signaling pathways require cell-cell contact. Notch ligands are mostly transmembrane proteins, particularly Jagged and Delta [59]. A bidirectional conversion between CSCs and non-CSCs can be triggered by an inflammatory stroma, which is characterized by elevated NF-κB signaling, enhancing Wnt activation, and inducing dedifferentiation of non-CSCs to CSCs [51]. Thus, the microenvironment seems to be crucial to maintain the properties of CSCs, preserve their phenotypic plasticity, and protect them from the immune system.

## 6. Metastatic Cancer Stem Cells

Metastatic cancer stem cells have the properties of cancer stem cells and ability to invade leading to metastasis. Both CSCs and cancer cells have the ability to metastasizes [60]. Genome sequencing studies suggest that primary tumors accumulate most of the mutations vital for metastasis, showing similarity between metastatic CSCs and primary CSCs [61,62]. In breast cancer, previous studies have shown CSCs identified by CD44 expression are involved in metastasis [54]. It was also shown that cancer cells expressing stem cell markers detected in the blood of breast cancer patients, when inoculated into immunodeficient mice, induced bone, liver, and lung metastasis [63]. In pancreatic cancer, only CD133^+^CXCR4^+^ cells, but not CD133^+^CXCR^-^ cells, demonstrated metastatic activity, even though both subsets have tumor-promoting capacity [29]. In colorectal cancers, metastasis was mostly seen in CSCs that exhibited long-term self-renewing capacity [64]. Multiple circulating CSCs home to the bone marrow and spawn metastases, suggesting that metastatic CSCs enter the circulation. The metastatic CSC may have evolved from the primary tumor CSC or from a non-CSC within the tumor by changes induced by niches, EMT changes, etc. [65]. Interestingly, dormancy plays a major role tumor recurrence and metastatic spread after long lag periods in many cancers, including breast, melanoma and leukemia [66,67,68]. Since dormant cells are proliferatively quiescent, they survive chemotherapy, radiation therapy better than proliferating cells and the surviving cells eventually regrowth. It is possible that restricted supplies of nutrients and oxygen due to poor vascularization cause growth arrest and dormancy. Circulating metastatic cells co-express EMT and stem cell markers [69]. Even though EMT enables migration, it interferes with proliferation and metastatic growth [70]. Thus, metastatic cells that have undergone EMT may need to reacquire an epithelial phenotype by mesenchymal-to-epithelial transition (MET) to seed and resume growth at a metastatic site [71]. Little is known about entering and exiting dormancy at present, so better models are needed to understand dormancy. Some studies have shown multiple CSC pools exist within individual tumors in ovarian, breast, and squamous-cell carcinomas. In these tumors, distinct CSC populations regenerate the phenotypic and functional heterogeneity of the parental tumor (Figure 1) [27,72,73]. In primary colorectal cancers, three different types of tumor-initiating cells were identified from single patient’s tumor. These tumor-initiating cells CSCs maintained tumor growth on serial transplantation, CSCs that are dormant or has partial self-renewal capacity, are activated in secondary or tertiary transplantation assays [64]. Tumor-initiating cells from PTEN-deficient glioblastoma, showed clonal heterogeneity, in which a series of phenotypically distinct self-renewing cells was observed in both the CD133^+^ and CD133^−^ fractions [74]. In skin squamous cell carcinoma, two CSC subsets located along the tumor-stromal interface displayed different growth kinetics and could interchange phenotypes [72]. The delineation of functionally distinct pools of CSCs requires cell-tracing studies in vivo in the future.

## 7. Targeting Cancer Stem Cell

Generally, CSCs appear to be resistant to conventional cancer therapies such as chemotherapy and radiation [42,75]. There is pre-clinical and clinical evidence to show CSCs are resistant to chemotherapy in a number of solid tumors, including breast [42,76]. A number of different strategies are explored to target CSCs, including self-renewal pathways, quiescence, CSC-specific cell surface molecules, stem cell niche, etc. There are many therapeutic agents against CSCs and are actively being evaluated in pre-clinical and clinical settings of various cancers [77]. Differentiation therapies that specifically targeted CSCs by exploiting their ability to differentiate can be effective is some cases. This strategy is successful in inducing cell-cycle progression in acute myeloid leukemia stem cells by G-CSF to promote sensitivity to chemotherapy [68]. Similarly, mouse glioblastoma stem cells when treated with BMP4 protein differentiates into Gila, resulting in reduced tumor growth, and tumor initiation capacity of CSCs upon transplantation [78,79]. Tumor cell plasticity present is a challenge to develop CSC targeted cancer therapies; a therapeutic eradication of CSCs might be followed by their regeneration from non-CSCs within the tumor under treatment [80]. Targeted therapies against Wnt, Notch and Hh pathways that frequently are deregulated in CSCs resulted in marked reduction in tumorigenic potential [42,81,82,83,84,85]. It was shown that high expression of Hg ligands in stromal plays a major role in the maintenance of CSCs and their niche. Notch pathway inhibition by a neutralizing antibody against the DLL4 ligand was effective in reducing CSC numbers in diverse solid xenograft tumors [86], whereas inhibition of Notch-4 expression within the CSC largely ablated breast tumor growth [87]. Tumor microenvironment supports CSCs, thus targeting CSC niche factors that regulate plasticity was shown to be effective in some tumors. Antibodies that abrogate the activation of c-Met by HGF significantly inhibited xenograft growth of colon tumors [86]. Antibodies against the fibronectin receptor α4β1 integrin reduced association of tumor cells with metastatic niches [88]. Targeting hypoxia through HIF-1α and HIF-2α, which promote cell cycle via Myc, was a promising therapy for glioma patients [89,90]. The biological activities of CSCs are regulated by several pluripotent transcription factors, such as OCT4, Sox2, Nanog, KLF4, and MYC [91] and these factors have become targets for cancer treatment [10]. Other therapeutic targets currently being pursued are MicroRNA Mir 34a that was shown to suppress breast and prostate cancer CSCs and metastasis [92,93]. MicroRNA 199a was shown to target CD44 to suppress the tumorigenicity and multidrug resistance of ovarian cancer-initiating cells CD44^+^/CD117^+^ [94]. In another study, miR145 was shown to inhibit CSCs by targeting Oct4 and Sox2 in GBM-CD 133^+^ and facilitated their differentiation into CD133^−^ non-CSCs [83].

## 8. Conclusions and Future Directions

An evolving consensus in the field is that under normal circumstances, normal stem cells (NSCs) maintain a homeostasis and replenish the adult cells while deregulation of NSCs can give rise to CSCs eventually leading to cancer and metastasis. The most studies show anti-CSC therapies to date could reduce rather than eradicated solid tumors in preclinical models. In standard clinical trials, tumor response criteria depend on measurements of tumor size, which largely reflects tumor responses to chemo- or radiation therapy in the non-CSC bulk tumor. Specific response criteria that provide a readout to anti-CSC agents in pre-clinical and clinical trials need to be standardized. An effective cancer therapy can be achieved by targeting the CSCs and bulk of tumor and creating a harsh microenvironment for the tumor cells to grow and metastasize.

## Figures and Tables

**Figure 1 genes-11-01372-f001:**
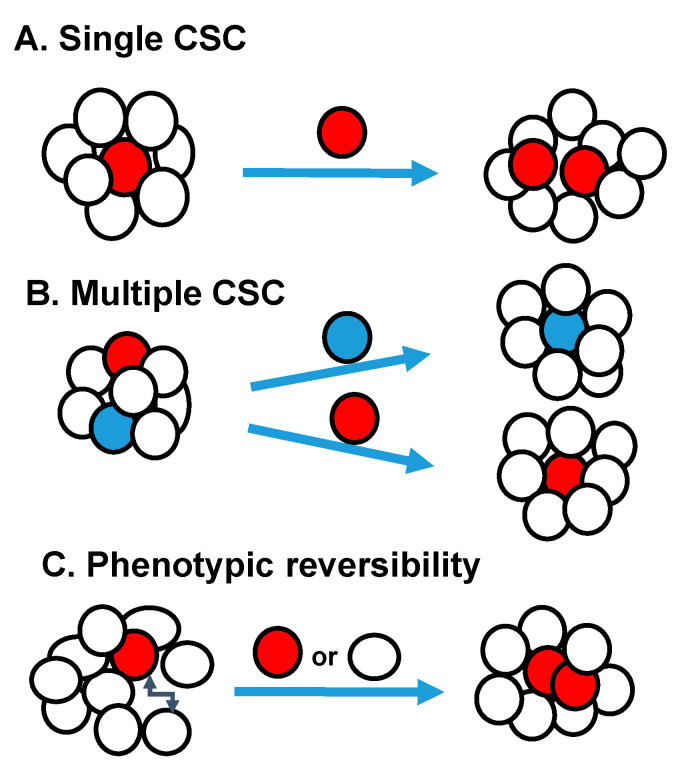
Schematic Models of Tumor Heterogeneity: (**A**) Single CSC pool may be present within the tumor. (**B**) Multiple distinct CSC pools with in individual tumors. (**C**) The CSC can be unstable, resulting in phenotypic reversion of cell-surface markers and CSC phenotype.

**Table 1 genes-11-01372-t001:** Distinguishing Characteristics of Normal and Cancer stem cells.

Normal Stem Cells (NSCs)	Cancer Stem Cells (CSCs)
Tightly regulated self-renewal capacity	Highly dysregulated self-renewal capacity
Generates normal progeny	Phenotypically diverse progeny
Normal Karyotype	Abnormal Karyotype
Relatively long telomeres	Short telomeres
Oxidative phosphorylation	Glycolysis
Normal oxygen through blood vessels	Highly resistant to lack of oxygen
Niche modifies local environment for immune protection of NSCs.	Niche modifies local environment for immune protection of CSCs
Niche maintains homeostasis	Deregulated niche promotes invasion and metastasis

**Table 2 genes-11-01372-t002:** CSCs markers in different cancers.

Type of Tumor	Cancer Stem Cell Markers
Acute Myeloid Leukemia	CD34^+^, CD38^+^
Breast	CD44^+^, CD24^−^, ALDH1^high^, CD133^+^
Colorectal	CD133^+^, CD44^+^, EpCAM^high^, ALDH1^high^
Glioblastoma	CD133^+^
Head and neck	CD44^+,^ CD24^+,^ ALDH1^high^, CD271
Liver	CD44^+^, CD90^+^, CD133^+^, ALDH1^high^
Lung	CD44^+^, CD90^+^, CD133^+^, ALDH1^high^
Ovarian	CD44^+,^ CD117^+^, CD133^+^
Skin	CD20^+^, CD271^+^
Melanoma	CD133^+^, ABCB5^+^, ALDH1^high^
Pancreatic	CD44^+^, CD24^+^, ESA^+,^ CD133^+^
Prostate	CD133^+^, CD44^+^, ALDH1^high^

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
