# Peer review of "Stem Cells: Current Status and Therapeutic Implications"

_genes, 2020, doi:10.3390/genes11111372_

Round 1
Reviewer 1 Report
English revision recommended.
Cancer stem cells do have a distinctive metabolic phenotype which escapes conventional therapies. Please add a discussion on this.
Please find the detailed comments in the attachment.

Author Response
please see the authors' reply in the attachment

Reviewer 2 Report
The review does not add anything to the literature on Cancer Stem cells. Some part appear to be written in hurry without any deeper consideration or explanation.
In the first paragraph on key features of normal and cancer stem cells, the Authors should expand the ability of both NSC and CSC to escape innate or adaptive immune system ( see also Pastò A et al IJMS, 2020). The sentence as such is not clear: the behavior of Normal stem cells is different from the one of cancer stem cells, since the immune microenvironment in which they reside is completely different. Explain in details, also in the paragraph of Immunological characteristics of cancer stem cells.
Please add clarification also to the sentence “quiescent most of the time”.
Please add reference to the abnormal karyotype and mitotically less active of CSC.
The whole structure of the review is difficult to follow: in the cancer stem cell’s niche the Authors describe immunological characteristics of cancer stem cells and the surrounding microenvironment that could be put together with the other aspect of CSCs described in the paragraph above.
Try to organize the structure of the review in a different manner.
A figure summarizing the similarities and differences between NSC and CSC should be added.
Figure 1 is not clear, it needed to be better explained in the text
Typing and grammar errors in the metastatic cancer stem cells paragraph.
Author Response

(The authors gave the same response as above.)

Round 2
Reviewer 2 Report
the Authors addressed all the points asked by the reviewer